# Highly Sensitive Detection for Mercury Ions Using Graphene Oxide (GO) Sensors

**DOI:** 10.3390/mi12091070

**Published:** 2021-09-02

**Authors:** Lei Liu, Haixia Shi, Raoqi Li, Cheng Liu, Jia Cheng, Li Gao

**Affiliations:** 1Department of Kidney Transplantation, The Second Xiangya Hospital of Central South University, Changsha 410011, China; liulei0831@csu.edu.cn; 2Physical Education Department, Jiangsu University, Zhenjiang 212013, China; shihaixia987@sina.com; 3School of Life Sciences, Jiangsu University, Zhenjiang 212013, China; raoqili2014@163.com (R.L.); jiangsuliu2021@163.com (C.L.); hbangxing@163.com (J.C.)

**Keywords:** graphene oxide, Hg^2^^+^ detection, DNA, sensors

## Abstract

The mercury ion (Hg^2+^) is one of the heavy metal ions, and its presence in trace amounts can cause physiological damage to an organism. Traditional methods of Hg^2+^ detection have been useful but have also had numerous limitations and challenges, and as a result, it is important to design new and sophisticated methods that can aid in the detection of Hg^2+^. In this paper, two fluorescent dyes, carboxyfluorescein (FAM) and SYBR Green I, were used to label and intercalate DNA probes immobilized on the surface of graphene oxide (GO) for sensors to detect Hg^2+^. FAM and SYBR Green I dye share close excitation and emission wavelength spectra, which can promote and amplify the detection of signals, and also increase the limit of detection (LOD). The results showed that the limit of detection in this method was 0.53 nM. Moreover, when the sensors with double amino groups on the surface of GO were carried out to detect Hg^2+^, a limit of detection was improved to 0.43 nM. The sensors were then applied in the real sample. The results show that this method has a promising potential in Hg^2+^ detection.

## 1. Introduction

Mercury ion (Hg^2+^) is one of the most toxic water pollutants, and is not only hazardous to the environment, but also harmful to human health. Excess accumulation of Hg^2+^ in humans can cause neurological disorders, bone softening and other medical conditions [1]. Trace amounts of Hg^2+^ in the human body can lead to irreversible lifelong tissue and organ injury and damage, causing headache, dizziness and fatigue, low fever, sleep disorders, emotional agitation and even death. Therefore, detecting Hg^2+^ has important implications for food safety and human health [2,3].

The United States Environmental Protection Agency (EPA) and the World Health Organization (WHO) have established strict regulations on the allowable Hg^2+^ concentration in drinking water to 1 μg/L (10 nM) [4]. To date, different types of analytical methods have been developed for Hg^2+^ detection with high sensitivity, such as colorimetric assays [5], fluorescence-based assays [6], inductively coupled plasma emission/mass spectrometry [7] and ion chromatography [8]. In 2004, Ono and Togashi found that Hg^2+^ specifically bound to DNA with multiple T probes to form a T-Hg^2+^-T structure with good selectivity [9,10], where the binding strength is even higher than that of base T-A. Due to the stability of DNA, the method of considering DNA as a recognition molecule is rapid, low cost, and suitable for real-time detection [11,12]. The methods for detecting Hg^2+^ based on DNA, and they have been developed rapidly which includes color reaction [13,14], electrochemistry [15,16] Raman [4,17,18], surface plasmon resonance [19,20], fluorescence intensity [21,22,23,24], etc. Fluorescence resonance energy transfer (FRET) [15,25,26,27] is also a technique widely used in the detection of Hg^2+^. However, the traditional methods have their limitations in terms of sensitivity or selectivity, and further, require the use of complicated instruments [28]. Huang et al. [29] used the DNA intercalating agent Ethidium Bromide (EB) and FAM as the fluorescent markers for the detection of SNPs. DNA was labeled and synergized with the post-adsorption quenching effect of GO. The limit of detection was reduced to 1 nM. One limitation to the method is the toxic and hazardous nature of EB, which we replaced with SYBR Green I, a non-toxic dye which can perform the same DNA intercalation function as EB. Sun et al. reported that the adsorption of the fluorescently labeled single-stranded DNA (ssDNA) probe on the carbon nanoparticle (CNP) via π–π stacking interactions between DNA bases and CNP leads to substantial dye fluorescence quenching. In the presence of Hg^2+^, T–Hg^2+^–T induced hairpin structure does not adsorb on CNP and thus retains the dye fluorescence [30]. Liu et al. reported that SYBR Green I efficiently discriminated against mercury-specific DNA and mercury-specific DNA/Hg^2+^ complex, which provided a label-free, fluorescence turned-on assay for Hg^2+^ detection. Qiu et al. further reported SYBR Green I dye was adsorbed on the surface of GO as a signal reporter for Hg^2+^ detection [31,32]. 

Covalent linking single-strand DNA to the surface of GO with a co-anchor strand improved the detection sensitivity of Hg^2+^ in our previous research [33]. In order to increase the limit of detection for Hg^2+^, the fluorescent dyes, FAM and SYBR Green I, combined with covalently linking single-strand DNA to the surface of GO with co-anchor strand, were used in this study. The excitation and emission wavelengths of the fluorescent dyes are close to each other, which can promote each other and further improve the sensitivity to Hg^2+^ detection.

## 2. Materials and Methods

### Reagents and Experiments

Multi-T probe 1 was 5′-TTT GCT TGT TGC GCT TCT TGC TTT-NH_2_-3′, Multi-T probe 2 was 5′-NH_2_-GAT AGC TTT GCT TGT TGC GCT TCT TGC TTT -FAM-3′, Multi-T probe 3 was 3’-NH_2_-CTA TCG-5’. The probes were purchased from Sangon Biotech Company (Shanghai, China) and purified by HPLC. Mercuric nitrate (Hg(NO_3_)_2_) was purchased from Shandong West Asia Chemical Industry Co., Ltd. (Lin Yi, China) and zinc chloride (ZnCl_2_), Nickel Chloride (NiCl_2_), Potassium Chloride (KCl), Lead Acetate ((CH3COO)_2_Pb), Magnesium Chloride (MgCl_2_), and Manganese Chloride (MnCl_2_) were purchased from Suzhou Asia Pacific Chemical Glass Instrument Co., Ltd. (Suzhou, China) Cadmium (CdCl_2_), calcium chloride (CaCl_2_), cobalt chloride (CoCl_2_), ferrous sulfate (FeSO_4_), ferric chloride (FeCl_3_), sodium chloride (NaCl), chromium trichloride (CrCl_3_), barium chloride (BaCl_2_), silver chloride (AgCl) were purchased from Sinopharm Chemicals (Shanghai, China), N-hydroxysuccinimide (NHS) and 1-3-Dimethylaminopropyl)-3-ethylcarbodiimide hydrochloride (EDC) was purchased from Sigma Co., Ltd. (St. Louis, MO, USA) SYBR Gree+-n I was purchased from Xiamen Zhishan Biotechnology Co., Ltd. (Xiamen, China) The Synergy H4 Multiplate Reader was used for fluorescence detection and Origin 8.0 was used for data analysis.

## 3. Preparation of Sensors

The modified Hummers method [34] was used to prepare GO from graphite powder. Ultrapure water was ultrasonicated for 20 min (1000 W) and then a uniformly dispersed GO mixture was obtained. The ultrapure water solution contains a mixture of 50 mM NHS and 200 mM EDC, and was mixed in a volume ratio of NHS/EDC mixture: ultrapure water: 1 mg/mL GO = 1:1:2, and allowed to stand at room temperature for 0.5 h. The mixture was then centrifuged at 10,000 rpm for 20 min to get the activated GO. 

Equal concentrations of probe 1 and probe 2 were mixed and placed in a water bath at 95 °C for 5 min for denaturing. Then the two probes were obtained with a partially double-stranded DNA probe 4 at room temperature for 2 h:

NH_2_-GAT AGC TTT GCT TGT TGC GCT TCT TGC TTT–FAM 

NH_2_-CTA TCG.

## 4. Detection of Hg^2+^

### 4.1. Amino-Immobilized DNA Sensors for Hg^2+^ Detection

An amount of 10 mM SYBR Green I was added to 200 μL of 50 μM multi-T probe 1 solution, then 10 μg/mL activated GO was added and allowed to react at 4 °C for 12 h. The sample was then removed and centrifuged at 10,000 rpm for 20 min to remove unfixed DNA probe. Different concentrations of Hg^2+^ were added to the homogeneous dispersion system for measuring the fluorescence intensity after 20 min. The excitation wavelength was set to 488 nm, the emission wavelength range was 510–650 nm, and the step size was 2 nm.

### 4.2. Amino-Immobilized DNA Sensors for Hg^2+^ Detection Combined with SYBR Green I

An amount of 10 mM SYBR Green I was added to 200 μL 50 nM partial double-stranded multi-T probe 4 solution and the FAM-modified DNA was intercalated with SYBR Green I. Then 10 μg/mL activated GO was added, and reacted at 4 °C for 12 h, then centrifuged at 10,000 rpm for 20 min to remove the excess DNA. Finally, different concentrations of Hg^2+^ were added to the system and the fluorescence intensity was measured after 20 min. The excitation wavelength was set to 488 nm, the emission wavelength range was 510–650 nm, and the step size was 2 nm.

### 4.3. The Optimization of SYBR Green l Concentration

As shown in Appendix A, the amino-labeled single-stranded nucleic acid probe was used, followed by labeling the sequence again with SYBR Green I. The probe was immobilized on the surface of activated GO to construct a dual fluorescent-group sensor for Hg^2+^ detection. Before carrying out these, the conditions of the experiments needed to be optimized. 

Amounts of 5 mM, 10 mM, 20 mM, and 40 mM of SYBR Green I fluorescent dyes were added to 50 nM DNA probe 1 solution for fluorescence detection after 10 min. Different concentrations of fluorescent dyes in the black histograms had different abilities to label DNA probes, among which 10 mM and 20 mM dyes had higher fluorescence intensity. The fluorescence was quenched by 10 μg/mL GO. The fluorescence detection was carried out after 20 min. If the fluorescence dye concentration was higher, the rate of intensity also increased. In this study, 10 mM SYBR Green I was selected and the quenching rate was over 80% (Appendix A). 

### 4.4. The Optimization of DNA Concentration

In this study, single-stranded DNA probe 1 acted as the molecular recognition moiety that interacted with Hg^2+^. Therefore, the choice of DNA concentration was important. In order to optimize the DNA concentration in the system, the experiments were carried out with different concentrations of DNA probes, and 10 μg/mL of activated GO was added to DNA solutions of different concentrations (10 nM, 30 nM, 50 nM) at 4 °C for 12 h. The samples were centrifuged at 10,000 rpm for 20 min to remove the excess DNA probe. When the same concentration of Hg^2+^ was added to the different systems, the fluorescence intensity recovery was linear with Hg^2+^ concentration after adding 10 mM SYBR Green I for 10 min at room temperature. The linear equations were:10 nM DNA: y = 69.168x + 2018.1
30 nM DNA: y = 278.44x + 3215.7
50 nM DNA: y = 525.75x + 5582.6

The fluorescence intensity increased with increasing DNA concentration. This showed that increasing DNA concentration can effectively improve the binding efficiency between DNA and activated GO. Moreover, according to our laboratory research report [35], a DNA concentration of 50 nM was the optimal concentration for the reaction (Appendix A).

## 5. Results

### 5.1. Experimental Design

As shown in Figure 1, the single-stranded nucleic acid probe labeled with FAM was used, followed by relabeling the probe with SYBR Green I. Thus, the probe was labeled with two dyes. The complementary DNA probe modified with an amino group was added. The single-stranded DNA probe formed the double-stranded probe in part and was modified with two amino groups. This improved the efficiency of the DNA probe on the surface of GO and further improved the detection sensitivity. The probe was immobilized on the surface of activated GO to construct a dual fluorescent-group sensor. After the addition of Hg^2+^, specific probes with enriched thymidine were induced to mismatch and form a hairpin structure. Then the interaction between GO and DNA was changed. The adsorption of the DNA probe on the surface of GO after the conformational change was weakened. Although one end was tightly bound, the probe as a whole was no longer bound to the surface of the GO. Therefore, the fluorescence intensity of the FAM, modified at the other end, was partially restored. The fluorescence intensity of SYBR Green I bound to the entire probe also recovered, and SYBR Green I had higher double-stranded binding efficiency than single-stranded, the hairpin structure itself increased the fluorescence intensity. The fluorescence of FAM and SYBR Green I was restored at the same time. 

### 5.2. The Optimization of GO Concentration

The 50 nM DNA probe 1 was added to different concentrations of activated GO and reacted overnight at 4 °C. The amino groups at the DNA end interacted with the carboxyl groups on the GO surface. Different concentrations of GO (2 μg/mL, 4 μg/mL, 6 μg/mL, 8 μg/mL, 10 μg/mL, 12 μg/mL, 14 μg/mL) and 50 nM DNA were mixed for overnight. After centrifugation, they were intercalated with 10 mM SYBR Green I, and then Hg^2+^ was added. Due to the quenching effect of GO, sensors were constructed with different concentrations of GO and did not show any significant changes after centrifugation. However, the higher the concentration of GO, with the addition of the same concentration of Hg^2+^, the more significant the increase in the fluorescence intensity. When the concentration reached 10 μg/mL, the corresponding fluorescence growth rate changed slowly (Appendix A). Therefore, this concentration was selected for this experiment.

### 5.3. Sensitive Detection

Prior to this experiment, Hg^2+^ was also detected by the physical adsorption of DNA. Probe 2 was used. A DNA probe of the same concentration of 50 nM was mixed with GO, and the two were combined by physical adsorption. After the addition of Hg^2+^, the fluorescence intensity increased with the addition of Hg^2+^. The limit of detection (LOD) was 22.5 nM (Appendix A). The results for the detection of Hg^2+^ using the immobilized DNA were shown in Figure 1A. After DNA was labeled with dye, energy transfer occurred between GO and immobilized single-stranded DNA, which caused the fluorescence of the dye intercalated into DNA to be quenched. The addition of Hg^2+^ binding to the DNA probe caused a conformational change and gradual recovery of fluorescence. As shown in Figure 1C, the fluorescence growth rate was linearly related to the Hg^2+^ concentration. The equation was *y* = 6.7134*x* − 0.0658, *R*^2^ = 0.9513, and the LOD based on 3S/N was 17.1 nM. Therefore, the detected minimum concentration of Hg^2+^ was important to the LOD. This method could reduce the LOD and increase the sensitivity compared with the physical adsorption.

### 5.4. Construction of FAM-Labeled Single-Stranded DNA Combined with SYBR Green I to Detect Hg^2+^

The experimental procedure was the same as that of the previous dual-fluorescent labeling DNA probe 2 sensor detection experiment. Before adding different concentrations of Hg^2+^ for 20 min, 10 mM SRBY Green I was added and mixed for 10 min. The results are shown in Figure 2A. GO interacted with single-stranded DNA and quenched the fluorescence of the dye bound to the DNA. After adding Hg^2+^, DNA-bound Hg^2+^ and fluorescence were recovered. As shown in Figure 2B, this was a linear correlation between the relative fluorescence intensity and Hg^2+^ concentration. The equation was *y* = 4.7221*x* + 0.0414, *R*^2^ = 0.9825. The LOD based on 3S/N was 0.53 nM. This result showed that the LOD was improved after the sensing system was constructed by GO-ssDNA labeled with FAM and intercalated with SYBR Green I.

### 5.5. Construction of a FAM-Labeled Double-Stranded Amino-Immobilized DNA Combined with SYBR Green I to Detect Hg^2+^

#### 5.5.1. Sensitive Detection

The experimental procedure was the same as that of the previous dual-fluorescent DNA sensor detection experiment. DNA would form a hairpin structure upon binding to Hg^2+^. Hg^2+^ and SYBR Green I inserted into DNA probe with FAM. This dsDNA included a double-amino group at the end. The fluorescence intensity of the entire sensor at different times after adding Hg^2+^ was shown in Appendix A. The intensity was stable in 20 min. Therefore, 20 min was chosen for the detection. The results are shown in Figure 3A. GO interacted with double-stranded DNA to quench the fluorescence of the dye bound to the probe. After Hg^2+^ was added, the conformation of the DNA probe was changed. Fluorescence was gradually recovered from the surface of GO. As shown in Figure 3B, the relative fluorescence intensity was linearly related to the Hg^2+^ concentration. The equation was *y* = 3.5043*x* + 0.0136, *R*^2^ = 0.9701. The LOD based on 3S/N was 0.43 nM. The LOD of Hg^2+^ observed in this study was comparatively better than that obtained from other assays (as shown in Table 1). This LOD was lower than some reported results. For example, Ge et al. [21] used a multi-T probe with G in combination with thioflavin T(ThT) to detect Hg^2+^ with a detection limit of 5 nM. Chiang et al. [22] reported that the detection limit was 3 nM when the fluorescent dye TOTO-3 was combined with a randomly rotated T33. Zhou et al. [23] used 2-aminopurine (2AP) and Zhu et al. [24] inserted DAPI (4,6-diamidino-2-phenylindol) into double-stranded DNA, and detected Hg^2+^ based on changes in fluorescence intensity. The detection limits were 3 nM and 1.5 nM. This showed that the method had a higher sensitivity. The methods with lower LOD than these methods were reported, such as Tu, where it was reported that the LOD was 40 pM using a liquid-gated graphene field-effect transistor (FET) [36]. Guo et al. reported a fluorescent sensor for Hg^2+^ detection and LOD was 0.17 nM using the acridine orange (AO) [37]. However, this method had lower LOD than most of the reported methods. Furthermore, some methods, such as Tu’s graphene FET, were more complicated than this method, although its LOD was 40 pM.

#### 5.5.2. Selective Detection

After the addition of Hg^2+^, specific probes with enriched thymidine were induced to mismatch and form a hairpin structure. Then, 1 μM of the same concentration of Hg^2+^ and other metal ions were added into the reaction mixture. Except for the different metal ion species, the other conditions were the same in the experiment. After 20 min at room temperature, their fluorescence intensities were detected. However, other metal ions can not bind to specific probes with enriched thymidine and form a hairpin structure. As shown in Figure 4, the effect of other metal ions on the fluorescence intensity was weaker. Hg^2+^ caused a significant increase in fluorescence intensity. The detection of Hg^2+^ using the sensor could be distinguished.

#### 5.5.3. Determination of Hg^2+^ in Real Samples

In order to further investigate the practical application potential of building sensors, Hg^2+^ detection was performed in tap water samples. To evaluate the reliability of sensor applications, repeated tests were performed at different concentrations (10 nM, 100 nM, and 1000 nM). The results are shown in Table 2. The recovery rate was from 89% to 110%. The results show that the sensor was more reliable in actual tests. The sensor constructed by this method had higher selectivity, sensitivity and reliability for the detection of Hg^2+^. The results of this test showed that no Hg^2+^ was detected in the tap water samples.

## 6. Discussions and Conclusions

The adsorbed DNA probes on the surface of GO are susceptible to non-specific displacement. This can lead to false-positive results. Therefore, the covalent method on the surface of GO was developed. The covalent sensors are much more stable and resistant to non-specific probe displacement. One fluorescent dye was used for most GO-DNA sensors. In this research, we studied the detection of Hg^2+^ by a single-stranded DNA probe that was covalent to the surface of GO with SYBR Green I. The LOD from DNA labeled with SYBR Green I was not lower than that of the DNA probe with FAM constructing sensors. SYBR Green I was a dye. Two fluorescent dyes can interact with each other. The excitation wavelengths of the two fluorescent dyes, FAM and SYBR Green I, amplified the fluorescence detection signals. This improved the sensitivity of detection. This sensor was constructed with a single amino-modified probe that had a detection limit of 0.53 nM. There was a four-fold reduction in LOD compared with the FAM alone. This showed that this method is promising with two fluorescence dyes. The LOD of the sensor was further constructed using the double-stranded DNA probe with the double-amino group. This can improve the covalent efficiency of the DNA probe on the surface of GO, which further reduced the LOD. The LOD was 0.43 nM for this method. Compared with other reported structures, this method had higher detection sensitivity and selectivity for Hg^2+^, and preliminary development in actual water samples. The results show that the sensor has good application prospects and is expected to be applied.

## Data Availability

Not applicable.

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
