# Peer review of "Highly Sensitive Detection for Mercury Ions Using Graphene Oxide (GO) Sensors"

_micromachines, 2021, doi:10.3390/mi12091070_

Round 1

Reviewer 1 Report

The authors of the article studied the detection of Hg2+ using fluorescent dyes, DNA probes, and GO. The paper although clearly presented lacks novelty. The manuscript should either be rejected or extensively revised.

  1. Since the focus of the work is more on the materials and their sensing properties rather than the device itself. The work does not fit into this journal. The paper is more suitable for the Sensors journal (MDPI) Sensors | An Open Access Journal from MDPI.
  2. (Abstract) Authors have mentioned “sensitivity of detection”, should be a limit of detection.
  3. In the introduction (lines 48-52) Authors have mentioned that they have used SYBR Green I dye instead of Ethidium Bromide which is toxic. But SYBR Green I and FAM dyes have been already used for the detection of Hg2+ [1,2]. In addition, SYBR green I dye and GO have also been used for the detection of Hg2+ [3]. The introduction part needs to be drastically improved, the relevant recent works and authors' contribution in this work need to be distinguished.   
  • Wang, Jing, and Bin Liu. "Highly sensitive and selective detection of Hg2+ in aqueous solution with mercury-specific DNA and Sybr Green I." Chemical communications39 (2008): 4759-4761.
  • Li, Hailong, et al. "Carbon nanoparticle for highly sensitive and selective fluorescent detection of mercury (II) ion in aqueous solution." Biosensors and Bioelectronics12 (2011): 4656-4660.
  • Qiu, Huazhang, et al. "A robust and versatile signal-on fluorescence sensing strategy based on SYBR Green I dye and graphene oxide." International journal of nanomedicine10 (2015): 147.
  1. The interaction between different materials needs to be studied using XPS or other material characterization tools.
  2. It is not clear how do the authors claimed that 50 nM of DNA concentration is optimal, although there is no data higher than 50nM of DNA concentration. 

  1. In most cases, the focus has been made to detect heavy metals at lower concentrations using low-cost devices. Thus, optimization should be carried at the concentration below the WHO permissible limit in the drinking water system. Figure SI 4 shows that 14 μg/mL GO concentration works better for both 1µM and 2µM concentrations of Hg2+.   What is the reason for using 10 μg/mL GO concentration?

  1. The authors should describe the mechanism of the selective detection of Hg2+ in detail.

Reviewer 2 Report

The authors submitted a manuscript about results obtained in the development of a sensor for a highly sensitive detection of Hg2+ using Graphene Oxide (GO).

  1. in the section 5.2.2 (optimization of DNA concentration) the slope increase with DNA concentration but R2 in two cases is lower than 0.9. How authors explain these Rvalues?
  2. Regarding calibration curves, linearity not always is as described. Please check data and figures.
  3. In Table 2 (Results for the determination of Hg2+ in the tap water (n=10)), what is the error calculated for Hg2+ measured in real samples?

Reviewer 3 Report

1. There are no articles in the literature review for 2019 - 2021. The authors should take into account the world experience of the last 3 years and compare the results of their research with it.
2. The quality of the drawings is rather low, sometimes the inscriptions are difficult to distinguish. The presentation of the results also require improvement.
3. It is advisable to single out the discussions in a separate paragraph and summarize all the data obtained in it. It would be helpful for the readers to see more physical interpretation of the results.
4. What is the reproducibility of the results? How many samples were taken and measurements were done?

Author Response

English was revised in this paper by my US friend, King Solomon.

Round 2

Reviewer 1 Report

Comment 1: The authors intentionally deleted one of the comments. This is unethical to remove the comment and not to address them.  Here is the deleted comment.

  1. Since the focus of the work is more on the materials and their sensing properties rather than the device itself. The work does not fit into this journal. The paper is more suitable for the Sensors journal (MDPI) Sensors | An Open Access Journal from MDPI.

If authors thought that the manuscript is appropriate to this journal, they should have given justification instead of removing the comment.

Comment 2 “Covalent linking single-strand DNA to the surface of GO with a co-anchor strand can improve the detection sensitivity of Hg2+ in our previous research.”

If this is from the published, authors should provide a reference.

Comment: In the introduction (lines 48-52) Authors have mentioned that they have used SYBR Green I dye instead of Ethidium Bromide which is toxic. But SYBR Green I and FAM dyes have been already used for the detection of Hg2+ [1,2]. In addition, SYBR green I dye and GO have also been used for the detection of Hg2+ [3]. The introduction part needs to be drastically improved, the relevant recent works and authors' contribution in this work need to be distinguished.   

  • Wang, Jing, and Bin Liu. "Highly sensitive and selective detection of Hg2+ in aqueous solution with mercury-specific DNA and Sybr Green I." Chemical communications39 (2008): 4759-4761.
  • Li, Hailong, et al. "Carbon nanoparticle for highly sensitive and selective fluorescent detection of mercury (II) ion in aqueous solution." Biosensors and Bioelectronics12 (2011): 4656-4660.
  • Qiu, Huazhang, et al. "A robust and versatile signal-on fluorescence sensing strategy based on SYBR Green I dye and graphene oxide." International journal of nanomedicine10 (2015): 147.

Response: We thank the referee for the suggestion. The following sentences were added in the introduction. “Sun et al. reported that the adsorption of the fluorescently labeled single-stranded DNA (ssDNA) probe on the carbon nanoparticle (CNP) via π–π stacking interactions between DNA bases and CNP leads to substantial dye fluorescence quenching. In the presence of Hg2+, T–Hg2+–T induced hairpin structure does not adsorb on CNP and thus retains the dye fluorescence[30]. Liu et al. reported that Sybr Green I efficiently discriminates mercury-specific DNA and mercury-specific DNA/Hg2+ complex,which provided a label-free, fluorescence turn on assay for Hg2+ detection. Qiu et al. further reported SYBR Green I dye was adsorbed on the surface of graphene oxide as signal reporter for Hg2+ detection[31-32].” “Covalent linking single-strand DNA to the surface of GO with a co-anchor strand can improve the detection sensitivity of Hg2+ in our previous research. In order to increase the limit of detection for Hg2+, the fluorescent dyes, FAM and SYBR Green I, combed with covalently linking single-strand DNA to the surface of GO with co-anchor strand were used in this study.” in the introduction distinguished the relevant recent works and authors' contribution in this work.   

Comment 3: The laboratory data authors have mentions should be included in the revised manuscript?

Comment: It is not clear how do the authors claimed that 50 nM of DNA concentration is optimal, although there is no data higher than 50 nM of DNA concentration.

Response: We thank the referee for the suggestion.The fluorescence intensity increased with increasing DNA concentration. This showed that increasing DNA concentration can effectively improve the binding efficiency between DNA and activated GO. Moreover, according to our laboratory research report, a DNA concentration of 50 nM was the optimal concentration for the reaction. Higher concentration of DNA is not necessary.

Authors have addressed other comments. The manuscript can be addressed after minor modifications.

Author Response

English was revised by my friend, King Solomon. Thanks.

Reviewer 2 Report

Respectable authors, 

The manuscript has been improved. The referee suggests a last check before submit it.

Author Response

English was revised by my friend, King Solomon.

Reviewer 3 Report

The results of the article are quite interesting, but it would be useful for the readers to see more discussion and physical interpretation, the paragraph on "Discussions and conclusion" is too short. The article after Results breaks off and there is a feeling of incompleteness. Paragraphs 3 and 4 would look more organic as the "Materials and Methods" subsections.

Author Response

Thank you for your comments.
